

# Mediterranean swordfish (*Xiphias gladius* Linnaeus, 1758) population structure revealed by microsatellite DNA: genetic diversity masked by population mixing in shared areas

Tommaso Righi[1], Andrea Splendiani[1], Tatiana Fioravanti[1], Andrea Petetta[1], Michela Candelma[1], Giorgia Gioacchini[1], Kyle Gillespie[2], Alex Hanke[2], Oliana Carnevali[1] and Vincenzo Caputo Barucchi[1]

[1] Dipartimento di Scienze della Vita e dell'Ambiente, Università Politecnica delle Marche, Ancona, Italy
[2] Fisheries and Oceans Canada, St. Andrews Biological Station, Ottawa, Canada

## ABSTRACT

**Background**. The Mediterranean swordfish stock is overfished and considered not correctly managed. Elucidating the patterns of the Mediterranean swordfish population structure constitutes an essential prerequisite for effective management of this fishery resource. To date, few studies have investigated intra-Mediterranean swordfish population structure, and their conclusions are controversial.

**Methods**. A panel of 20 microsatellites DNA was used to investigate fine-scale population structuring of swordfish from six main fishing areas of the Mediterranean Sea.

**Results**. This study provides evidence to reject the hypothesis of a single swordfish population within the Mediterranean Sea. DAPC analysis revealed the presence of three genetic clusters and a high level of admixture within the Mediterranean Sea. Genetic structure was supported by significant $F_{ST}$ values while mixing was endorsed by the heterozygosity deficit observed in sampling localities indicative of a possible Wahlund effect, by sampling admixture individuals. Overall, our tests reject the hypothesis of a single swordfish population within the Mediterranean Sea. Homing towards the Mediterranean breeding areas may have generated a weak degree of genetic differentiation between populations even at the intra-basin scale.

# INTRODUCTION

Sustainable harvesting of fish stocks is one of the primary objectives of fishery management (*Cochrane, 2002*). Fish species frequently form different populations, which are genetically structured through behavioural and geographical distributional differences (*Reiss et al., 2009*). Consequently, the identification of population structure is an essential prerequisite for effective and sustainable management of the fishery resources (*Reiss et al., 2009*).

Corresponding author
Vincenzo Caputo Barucchi,
v.caputo@univpm.it

Delineation of spatial management areas ignoring the spatial distribution and the relationships between fish populations may result in depletion or extinction of the most vulnerable or local subpopulations and loss of genetic diversity that reduces the ability for the species to evolve and adapt to environmental changes. However, a frequent mismatch between ecological and biological processes and management actions in the last decade has led to the decline of many fish stocks (*Reiss et al., 2009*).

This is particularly true for marine pelagic fish species where the identification of population structure is hampered by the low level of intraspecific heterogeneity (*Ward, Woodwark & Skibinski, 1994*) as a result of the absence of geographical barriers, long larval periods and their widespread dispersal as well as highly migratory adults that facilitate high levels of gene flow. However, demographic history of the ancestral population combined with current ecological biogeographic factors, such as dispersal potential, spawning behaviour and population size, have led to population differentiation, as revealed by several genetic studies for different fish species (*Zardoya et al., 2004*; *Martínez et al., 2006*; *Ruzzante et al., 2006*; *O'Leary et al., 2007*; *Pecoraro et al., 2016*; *Pecoraro et al., 2018*; *Ruggeri et al., 2016a*; *Ruggeri et al., 2016b*).

Swordfish is a pelagic and highly migratory species, distributed worldwide from 45°N to 45°S in the open waters of the Atlantic, Indian, and Pacific oceans. Swordfish also occurs in the Mediterranean, Marmara, Black and Azov seas (*Palko, Beardsley & Richards, 1981*). The 2011 International Union for the Conservation of Nature (IUCN) assessment for this species has shown a 28% decline in total biomass over the last 20 years globally, and the Mediterranean stock is currently overfished and not well-managed (*Collette et al., 2011*). The 2016 International Commission for the Conservation of Atlantic Tunas (ICCAT) Mediterranean swordfish stock assessment (2016) reported that the Mediterranean swordfish spawning stock biomass (SSB) level is less than 15% of biomass at maximum sustainable yield, and between 50 and 70% of total yearly catches were represented by small-sized individuals (*International Commission for the Conservation of Atlantic Tunas, 2019*). Moreover, a recent study detected the loss of mitochondrial genetic diversity in mediterranean swordfish population during the last 20 years (*Righi et al., 2020*).

For management purposes, ICCAT considers populations in the Mediterranean Sea, North Atlantic, and South Atlantic populations as three separate stocks. Differentiation also occurs in biology with the Mediterranean swordfish having different characteristics compared to the Atlantic stocks such as a lower growth rate and younger age at sexual maturity than in the Atlantic populations (*Cavallaro, Potoschi & Cefali, 1991*; *Ehrhardt, 1992*; *Tserpes & Tsimenides, 1995*; *Arocha, 2007*). Moreover, genetic studies have confirmed genetic differentiation among the Atlantic and Mediterranean stocks (*Bremer, Baker & Mejuto, 1995*; *Kotoulas et al., 1995*; *Kotoulas et al., 2007*; *Bremer et al., 1996*; *Rosel & Block, 1996*; *Pujolar, Roldan & Pla, 2002*; *Viñas et al., 2007*). Separate analyses using mitochondrial and nuclear markers have highlighted a high level of genetic differentiation among the Atlantic and Mediterranean populations (see *Bremer et al., 2007* for a summary).

Differences among these basins were also observed in terms of discrete spawning areas, as well as different spawning periods (*Neilson et al., 2013*). In the North Atlantic, reproduction takes place in the western subtropical area, with year-round spawning and seasonal peaks
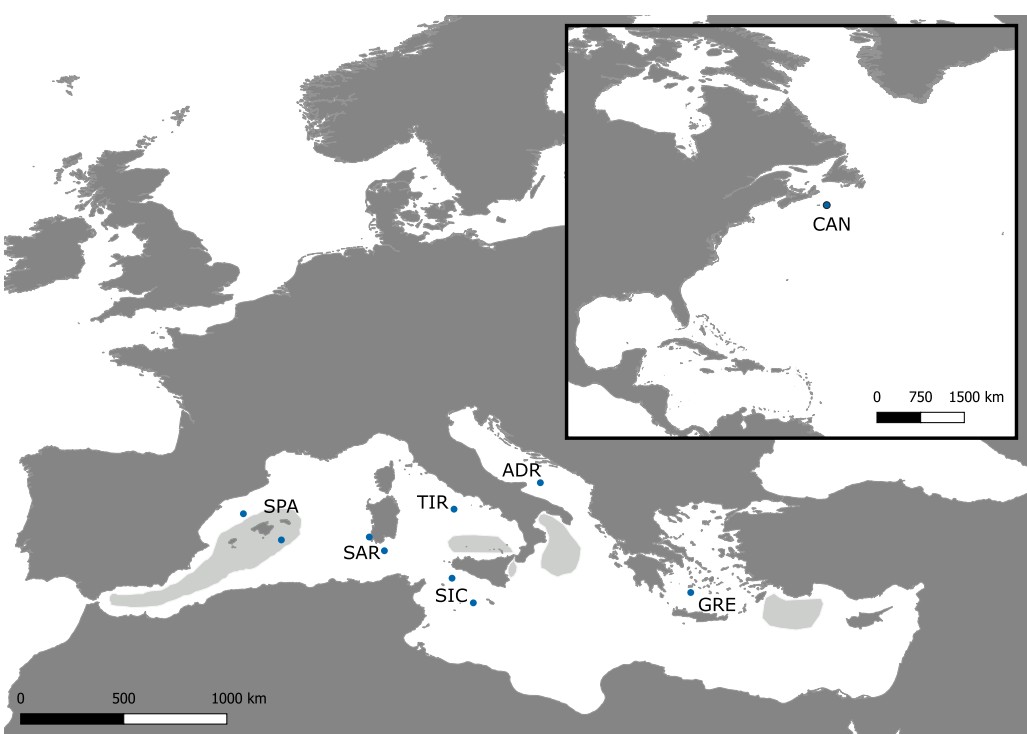

**Figure 1  Sampling map.** Map of the Mediterranean Sea and Northwestern Atlantic with locations of the Swordfish samples collected for the present study (dots). The approximate extents of the spawning areas are shown in light grey (*Arocha, 2007*).

(*Arocha, 2007*). In the Mediterranean Sea, spawning takes place between June and August (*Palko, Beardsley & Richards, 1981*; *Arocha, 2007*; *Marisaldi et al., 2020*) and it is restricted to three main spawning grounds. The first is located in the western Mediterranean, from the Strait of Gibraltar up to the Balearic Islands (*Arocha, 2007*; *Neilson et al., 2013*); the second one extends from the Strait of Messina to the Gulf of Taranto in the Ionian Sea (*Cavallaro, Potoschi & Cefali, 1991*; *Arocha, 2007*). The last one is close to Rhodes and Cyprus islands in the Levantine Sea (*Tserpes, Peristeraki & Somarakis, 2001*; *Tserpes, Peristeraki & Valavanis, 2008*) (Fig. 1). Philopatric behaviour has been proposed as the driving force behind global swordfish structuring and high levels of genetic differentiation between separate breeding areas, support the assumption of spawning site fidelity in the Atlantic Ocean (*Bremer et al., 2005a*). Moreover, no evidence of gene flow was observed using mtDNA between Mediterranean and North Atlantic swordfish populations despite mixing in the feeding area to the west of the Strait of Gibraltar (*Bremer et al., 2007*; *Viñas et al., 2007*).

While there are several studies on the global genetic structure of swordfish, few have focused on the Mediterranean swordfish stock. Some of these studies suggest the occurrence of a homogeneous stock in the Mediterranean Sea (RFLP of the entire mitochondrial DNA, (*Kotoulas et al., 1995*); RFLP of the mitochondrial control region and the nuclear Calmodulin gene, (*Chow & Takeyama, 2000*); four microsatellites, (*Kotoulas et al., 2007*). Conversely, *Viñas et al. (2007)* used the analysis of mtDNA CR-I sequences to propose

the existence of at least two distinct Mediterranean populations: one in the eastern basin and the other in the western basin. Although less evident, this pattern was detected using allozyme data (*Pujolar, Roldan & Pla, 2002*). These conflicting results regarding the occurrence of a genetic structure for the Mediterranean population of swordfish suggest the need for thorough genetic studies focusing on this basin to shed light on Mediterranean swordfish population structure and to provide reliable evidence for management actions. In the present study, a multi-locus approach based on the screening of a panel of 20 microsatellites was used to investigate the fine-scale genetic structure of swordfish within the Mediterranean Sea. Sampling coverage differs from all previous genetic studies by offering a more comprehensive representation of the Mediterranean area. In addition, the genetic differentiation between Atlantic and Mediterranean swordfish populations was investigated.

## MATERIALS & METHODS

### Sampling and DNA extraction

A total of 298 swordfish were collected from six areas within the eastern, central and western Mediterranean regions, to obtain a representative coverage of the basin. Twenty-five swordfish from the eastern coast of Canada have been included in the analysis as a comparison (For details see Table 1 and Fig. 1). Samples were collected at the fishing landing of the commercial catch by longline or trap bycatch (only in the case of Sardinian samples) from May to October in three years 2016–2018. Since sampling occurred on commercial landings, no young-of-the-year individuals (Lower jaw fork length (LJFL) = 60–70 cm) were collected. Fish LJFL ranged from 81 cm to 236 cm. For each sample, a piece of the caudal fin or muscle tissue was collected and stored in ethanol absolute and kept at $-20\,°C$ until DNA extraction. The procedures did not include animal experimentation, and ethics approval was not necessary following the Italian legislation (D.L. 4 of Mars 2014, n. 26, art. 2). Total genomic DNA was extracted using specific cartridge 401 in the *MagCore*® automated Nucleic Acid extractor (*MagCore*®, *Genomic DNA Tissue Kit, n° 401*) following the manufacturers' protocols.

### Microsatellite amplification, genotyping and diversity analysis

All specimens were genotyped for 20 microsatellite loci: three from *Muths et al. (2009)*: D2A, D2B, C8, eleven from *Kasapidis et al. (2009)*: XgSau98R1, Xgl-14, Xgl-35, Xgl-65b, Xgl-74, Xgl-94, Xgl-106, Xgl-121, Xgl-148b, Xgl-523b, Xgl-561) and six from *Reeb, Arcangeli & Block (2003)*: Xg-56, Xg-66, Xg-144, Xg-166, Xg-394, Xg-402). The PCR amplifications were performed combining microsatellite loci in five multiplex reactions based on primer's annealing temperature (details are shown in Table 2). The PCR amplification conditions consisted of: $1\times$ MyTaq Reaction buffer (Bioline) ($15\ \text{mmol L}^{-1}$ $MgCl_2$, $1,25\ \text{mmol L}^{-1}$ of each dNTP, plus stabilizers and enhancers), $0.3\ \mu M$ of each primer, 0.2 U Taq DNA polymerase (MyTaq, Bioline) and 40–80 ng of genomic DNA in a final volume of $10\ \mu L$ for 2- and 3-plex and $15\ \mu L$ for 4- and 7-plex, respectively. Each forward primer was labelled with a different fluorescent dye (FAM, VIC, NED, and PET) and set up avoiding the overlap of similar allele sizes. PCR conditions were optimized for all loci using the

**Table 1  Sampling details for swordfish analysed in this study.**

| Sampling area (FAO fishing area/Geographical subarea) | Sample ID | Sampling date | n | Size range (LJFL) |
|---|---|---|---|---|
| Balearic Sea (GSA 5, 6) | SPA | 07/16, 08/16, 09/16 09/18 | 85 | 90–199 cm |
| Southern Sicily (GSA 15, 16) | SIC | 07/16 06/17, 07/17 06/18 | 61 | 83–236 cm |
| Aegean Sea (GSA 22) | GRE | 08/16 | 20 | 97–167 cm |
| Southern Adriatic Sea (GSA 18) | ADR | 09/16 | 62 | 90–130 cm |
| Tyrrhenian Sea (GSA 10) | TIR | 05/17 | 16 | 90–170 cm |
| Sardinian Sea (GSA 11.2) | SAR | 06/17, 10/17 05/18, 06/18 | 54 | 81–204 cm |
| NW Atlantic (FAO Fishing area 21) | CAN | 08/18, 09/18, 10/18 | 25 | NA |
| Total | | | 323 | |

following touchdown protocol: initial denaturation at 95 °C for 5 min, followed by 10 cycles of denaturation at 92 °C for 20 s, annealing at the specific temperature (Ta) (Table 2) for 30 s, and extension at 72 °C for 45 s. At each cycle, the annealing temperature decreased by 0.5 °C. After that, samples were subjected to an additional 25 cycles of denaturation at 90 °C for 30 s, annealing at [Ta - 5 °C] for 50 s, and extension at 72 °C for 55 s. The reaction finished with a final elongation at 72 °C for 5 min. Amplified fragments were separated by electrophoresis using ABI Prism 3130xl genetic analyser executed by BMR-Genomics (Padua). Alleles were scored using GS 500LIZ_3130 size standard using Peakscanner 2 (Applied Biosystems), and the outputs were manually evaluated. The binning of alleles was accomplished using Flexibin 2 (*Amos et al., 2007*), to minimize microsatellite alleles miscalling.

Micro-Checker 2.2.3 (*Van Oosterhout, 2003*) was employed to test for genotyping errors due to stuttering, allelic dropout and the presence of null alleles. FreeNA (*Chapuis & Estoup, 2006*) was used to estimate if the presence of null alleles affected $F_{ST}$ estimation. Therefore, global $F_{ST}$ was calculated by both including null alleles (INA) and excluding null alleles (ENA). The bootstrap 95% confidence intervals (CI) for the global $F_{ST}$ values were calculated using 50,000 replicates over all loci. Allelic richness (*AR*), which is a standardised index of the mean number of alleles per locus irrespective of sample size, was estimated using Fstat 2.9.3 (*Goudet, 2001*). Observed ($H_O$) and expected ($H_E$) heterozygosity was computed with Arlequin v. 3.5.2.2 (*Excoffier & Lischer, 2010*). The Fisher's exact test was performed to evaluate deviation from Hardy–Weinberg equilibrium for each population and each locus using Genepop online software (*Raymond & Rousset, 1995*). The same software was also used to test linkage disequilibrium (LD) for all pairs of loci. Exact *P*-values were estimated using the Markov Chain algorithm (10,000 dememorization steps,
**Table 2  List of microsatellite loci used with corresponding repeat motif, fluorescent dye, annealing temperature, size ranges, and multiplex groups.**

| Locus | Repeat motif | Fluorescent dye | $T_a(°C)$ | Size range of alleles | Multiplex |
|---|---|---|---|---|---|
| Xgl-35 | $(CA)_{13}$ | NED | 58 | 196–251 | 1 |
| Xgl-121 | $(GT)_6(GC)_5(GT)_6$ | 6-FAM | 58 | 98–112 | 1 |
| Xgl-561 | $(CA)_6GA(CA)_7$ | VIC | 58 | 128–154 | 1 |
| Xgl-94 | $(GGA)_8$ | 6-FAM | 58 | 183–198 | 1 |
| Xgl-106 | $(GA)_{10}$ | PET | 58 | 203–239 | 1 |
| Xgl-65b | $(CT)_{16}$ | 6-FAM | 58 | 261–297 | 1 |
| Xgl-74 | $(AGG)_7$ | VIC | 58 | 232–250 | 1 |
| XgSau98R1 | $(CA)_8$ | NED | 58 | 155–187 | 2 |
| Xgl-523b | $(GA)_6AAGG(GA)_6GC(GA)_8$ | 6-FAM | 58 | 95–99 | 2 |
| Xgl-14 | $(CAT)_6CAC(CAT)_3\,CAC(CAT)_4(CGT)_7$ | 6-FAM | 58 | 143–186 | 2 |
| Xg-148b | $(GGA)_8$ | 6-FAM | 58 | 221–236 | 2 |
| D2A | $(CCT)_6$ | NED | 50 | 289–298 | 3 |
| D2B | $(CAGT)_8$ | PET | 50 | 161–191 | 3 |
| C8 | $(CTAT)_{22}$ | VIC | 50 | 143–247 | 3 |
| Xg-394 | $(TCC)_9$ | 6-FAM | 66 | 140–146 | 4 |
| Xg-402 | $(TCC)_5+(CTT)_2$ | 6-FAM | 66 | 186–194 | 4 |
| Xg-56 | $(CA)_{16}$ | NED | 53 | 117–148 | 5 |
| Xg-66 | $(CA)_{11}$ | PET | 53 | 126–144 | 5 |
| Xg-144 | $(GGA)_7$ | PET | 53 | 152–170 | 5 |
| Xg-166 | $(CAA)_7$ | VIC | 53 | 121–142 | 5 |

100 batches and 5,000 iterations) and the significance of HWE and LD values were adjusted with Bonferroni correction.

## Population structure analysis

Global $F_{ST}$ values (*Weir & Cockerham, 1984*) were analysed using Fstat 2.9.3. The 95% C.I. was estimated on 1000 iterations. Pairwise $F_{ST}$ distances among sampling localities were estimated in Arlequin 3.5.2.2 using 10,000 permutations ($p < 0.001$). Furthermore, both the Bayesian clustering method implemented in STRUCTURE 2.3.4 (*Pritchard, Stephens & Donnelly, 2000*) and the Discriminant Analysis of Principal Components (DAPC) (*Jombart, Devillard & Balloux, 2010*) were used to find the number of discrete genetic populations. The two clustering approaches were used in this study to compare the results. Different clustering approaches may lead to different conclusions (*Latch et al., 2006*; *Waples & Gaggiotti, 2006*; *Frantz et al., 2009*; *Kanno, Vokoun & Letcher, 2011*). DAPC, unlike STRUCTURE, does not rely on a specific population genetic model, and it is, therefore, free of assumptions about Hardy–Weinberg equilibrium or linkage disequilibrium. The absence of any assumption about the underlying population genetics model is one of the main assets of DAPC analysis, that results more suitable to unravel structuring in more complex population genetic models and may be more efficient at identifying genetic cline and hierarchical structure (*Jombart, Devillard & Balloux, 2010*). The Bayesian analysis (STRUCTURE), carried out by using the microsatellite dataset, was based on 10 consecutive

runs per cluster (K) where K ranged between 1 and 7. The admixture model and correlated allele frequencies model were set up. All analyses were run for $5 \times 10^5$ generations after a burn-in of $10 \times 10^4$ generations. The number of clusters that best fit the observed genotype data was determined comparing Delta K ($\Delta$K) (*Evanno, Regnaut & Goudet, 2005*) and the mean logarithmic probability of K, LnP(K), using the StructureSelector website (*Li & Liu, 2018*). The individual posterior probabilities of assignment (*q*) to specific cluster were estimated. The DAPC analysis was executed using package *adegenet* (*Jombart, 2008*). The firstly an explorative analysis was performed including apriori sampling geographical information to visualise the relationship between sampling locations. Cross-validation was used to select the number of principal components (PCs) to retain for DAPC. The lowest number of components for which the correct assignment probability stabilized was 40. Secondly, the optimal number of clusters (K) was evaluated with the function *find.cluster* which runs successive rounds of k-means clustering with an increasing number of clusters (K ranged from 1 to 5) and 500 runs at each value of K. Bayesian information criterion (BIC) was used to select the optimal number of clusters. Ideally, the lowest BIC value represents the optimal number of clusters, but, BIC values may keep decreasing after the true K value in the presence of genetic clines and hierarchical structure (*Jombart, Devillard & Balloux, 2010*). To choose the best number of K, the subdivision of individuals was investigated for K from 2 to 5. The analysis was performed both including and excluding the Canadian sample. Finally, pairwise $F_{ST}$ values among the best number of genetic clusters detected by DAPC were calculated in Arlequin.

## RESULTS

### Microsatellite genetic diversity

Twenty microsatellite loci were amplified for all 323 swordfish samples. The PCR failure per locus ranged between 0 and 5.9%, with an average of 1.1%.

Significant deviations from Hardy-Weinberg equilibrium were detected in 18 out of 140 single locus exact tests at loci Xgl-94, Xgl-74, Xgl-14, Xg–66 and Xg–166 (Table S1), considering sampling localities separately. Pooling together Mediterranean swordfish, 5 out of 20 loci deviated significantly from Hardy-Weinberg equilibrium (Xgl-94, Xgl-74, Xgl-14, Xg–66, Xgl-523). All deviations were towards a heterozygote deficit. Micro-checker showed evidence of null alleles (frequency > 0.3) for Xgl-74 and Xg-66. These loci deviated from Hardy-Weinberg equilibrium in many populations, including the Canadian samples and were, therefore, removed from further analysis. Rarer null alleles at Xgl-94, Xgl-14 and Xg –166 were detected. However, these latter loci were retained because the presence of null alleles did not bias the population differentiation parameters. In fact, the estimation of $F_{ST,}$ including and excluding the ENA correction method gave comparable results; $F_{ST} = 0.018$ with the respective 95% CI [0.011–0.027]. No consistent evidence for linkage disequilibrium was detected between pairs of loci within populations.

All the remaining 18 loci were polymorphic, with the number of alleles per locus ranging from two at locus Xg-402, to 21 at locus C8. Both, Xg-402 (two alleles) and Xg-394 (three alleles) loci were monomorphic in three samples GRE, SPA and CAN. The Mediterranean

**Table 3   Descriptive statistics for 18 microsatellite loci over the seven swordfish locations.**

| Marker | n | NA | MNA (s.d) | AR | $H_O$ (s.d) | $H_E$ (s.d) | HW disequilibrium | $F_{IS}$ (s.e) | Null alleles (s.d) |
|---|---|---|---|---|---|---|---|---|---|
| Xgl–35 | 322 | 14 | 6.143 (3.132) | 4.999 | 0.506 (0.150) | 0.517 (0.170) | n.s. | 0.002 (0.023) | 0.008 (0.020) |
| Xgl–121 | 323 | 8 | 4.429 (0.976) | 4.071 | 0.554 (0.061) | 0.583 (0.100) | n.s. | 0.043 (0.039) | 0.024 (0.036) |
| Xgl–561 | 323 | 9 | 6.429 (0.976) | 5.607 | 0.570 (0.114) | 0.588 (0.115) | n.s. | −0.005 (0.039) | 0.025 (0.036) |
| Xgl–94 | 321 | 6 | 5.143 (0.378) | 4.927 | 0.587 (0.113) | 0.740 (0.015) | Significant | 0.250 (0.053) | 0.086 (0.058) |
| Xgl–106 | 322 | 18 | 6.286 (3.861) | 5.371 | 0.700 (0.158) | 0.67 (0.119) | n.s. | 0.000 (0.042) | 0.012 (0.016) |
| Xgl–65b | 321 | 12 | 7.429 (1.718) | 6.669 | 0.657 (0.118) | 0.755 (0.066) | Significant | 0.111 (0.041) | 0.054 (0.019) |
| XgSau98R1 | 320 | 14 | 10.714 (1.380) | 9.337 | 0.846 (0.037) | 0.876 (0.022) | n.s. | 0.034 (0.027) | 0.013 (0.014) |
| Xgl–523 | 322 | 3 | 2.143 (0.378) | 2.090 | 0.310 (0.118) | 0.342 (0.086) | n.s. | 0.095 (0.084) | 0.245 (0.112) |
| Xgl–14 | 315 | 13 | 6.571 (2.573) | 5.211 | 0.449 (0.154) | 0.65 (0.110) | Significant | 0.341 (0.069) | 0.141 (0.066) |
| Xgl–148b | 323 | 4 | 3.143 (0.378) | 2.967 | 0.341 (0.081) | 0.343 (0.062) | n.s. | 0.070 (0.057) | 0.019 (0.033) |
| D2A | 322 | 4 | 3.143 (0.378) | 3.003 | 0.441 (0.149) | 0.442 (0.092) | n.s. | 0.040 (0.035) | 0.105 (0.053) |
| D2B | 322 | 7 | 3.857 (1.464) | 3.352 | 0.580 (0.103) | 0.624 (0.044) | n.s. | 0.066 (0.048) | 0.030 (0.040) |
| C8 | 319 | 21 | 12.714 (3.200) | 10.177 | 0.866 (0.061) | 0.873 (0.033) | n.s. | 0.002 (0.027) | 0.011 (0.025) |
| Xgl–394 | 319 | 3 | 1.571 (0.787) | 1.351 | 0.060 (0.040) | 0.025 (0.038) | n.s. | −0.031 (0.013) | 0.000 (0.000) |
| Xgl–402 | 321 | 2 | 1.429 (0.535) | 1.320 | 0.047 (0.021) | 0.020 (0.026) | n.s. | −0.017 (0.009) | 0.000 (0.000) |
| Xg–56 | 314 | 13 | 8.571 (2.225) | 8.150 | 0.780 (0.039) | 0.795 (0.050) | n.s. | 0.008 (0.026) | 0.020 (0.032) |
| Xg–144 | 317 | 7 | 4.000 (1.000) | 3.494 | 0.596 (0.072) | 0.674 (0.025) | Significant | 0.082 (0.047) | 0.049 (0.033) |
| Xg–166 | 308 | 8 | 6.429 (0.787) | 6.319 | 0.637 (0.141) | 0.740 (0.031) | Significant | 0.108 (0.065) | 0.121 (0.066) |

**Notes.**

n, number of individuals typed; NA, allele number; MNA, mean number of alleles; AR, allelic richness; $H_o$, observed heterozygosity; $H_E$, expected heterozygosity; HW, disequilibrium, significance for the Hardy–Weinberg disequilibrium after Bonferroni correction; $F_{IS}$, null allele frequencies; s.d. and s.e., standard deviation and standard error.

samples exhibited a significantly lower number of alleles per locus, allelic richness, and expected heterozygosity, compared to the Atlantic samples (Table 3 and Table S1). Instead, the mean number of alleles and their level of heterozygosity resulted similar among Mediterranean samples. Thus, no evidence of a geographical pattern was observed for the distribution of genetic variability among Mediterranean samples.

## Genetics structure

The global $F_{ST}$ showed a significant signal for genetic differentiation between Mediterranean and Atlantic populations ($F_{ST} = 0.091$; 95% CI [0.056–0.133]). The $F_{ST}$ values decreased when considering the Mediterranean samples separately ($F_{ST} = 0.018$; 95% CI [0.011–0.027]). Pairwise $F_{ST}$ across all samples ranged from 0 to 0.097. Low and statistically not significant values were detected between Mediterranean localities, whereas the higher and significant values ($p < 0.001$) were observed when comparing the Mediterranean and Atlantic samples, $F_{ST}$ values ranging from 0.083 (CAN-GRE) to 0.097 (CAN-SAR). The Bayesian clustering analysis suggested the presence of two genetic clusters ($K = 2$) using $\Delta K$ method, while LnP(K) showed an increase to $K = 2$ before declining and subsequently increasing to $K = 6$ (Figs. 2A–2B). For $K = 2$, individuals were partitioned according to their source basins (Mediterranean *vs* North-western Atlantic). Individuals were assigned to the specific cluster with a high score showing an average $q = 0.98$ for the Mediterranean group and an average $q = 0.95$ for the Atlantic group. Few individuals showed signs of

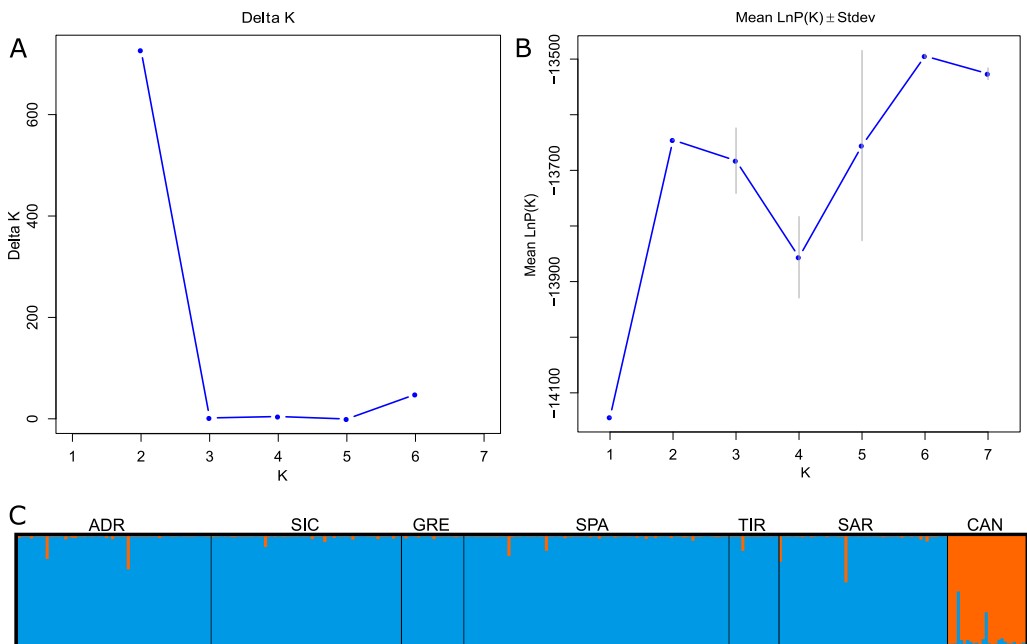

**Figure 2 Bayesian structure analysis implemented in STRUCTURE.** (A) Plot of Delta K and (B) Mean LnP(K) according to K. (C) Structure clustering results obtained at $K = 2$. Barplots showing posterior probabilities of swordfish individual genotypes (as bars) assigned to each population. The black lines separate sampling localities.

mixing between groups (Fig. 2C). Considering $K = 6$, Atlantic swordfish formed a single cluster, while all Mediterranean swordfish were uniformly assigned to the remaining five groups which failed in population structure identification (graph not shown).

The exploratory multivariate analysis, considering a priori sampling groups, divided swordfish into two groups. The first discriminant component (81% of variance) distinguished between Atlantic and Mediterranean swordfish, clustering all Mediterranean sampling localities together. This result is consistent with inter-oceanic genetic differentiation (Fig. 3).

Conversely, *a posteriori* assignment by DAPC analysis, despite the absence of a clear best value for the number of clusters, suggested the presence of more than two genetic clusters. The BIC graph showed an apparent decrease until $K = 4$, reaching a slightly lower value at $K = 5$ (Fig. 4A). For $K = 2$ and 3 high numbers of Mediterranean specimens were assigned into the Atlantic cluster and was in contrast with interoceanic differentiation reported in literature, STRUCTURE and explorative DAPC results (Figs. 4B, 4C, 4F). For $K = 4$, an evident interoceanic structure, in line with previous results, was detected (Fig. 4D). All Atlantic individuals but two clustered into a single group while Mediterranean swordfish were subdivided into the remaining three groups. Mediterranean groups were randomly distributed among sampling localities, and no geographic pattern was observed (Fig. 4F). Increasing the number of K, increased admixture and mixing within the Mediterranean samples. Therefore, $K = 4$ was selected as the possible optimal number of groups. The

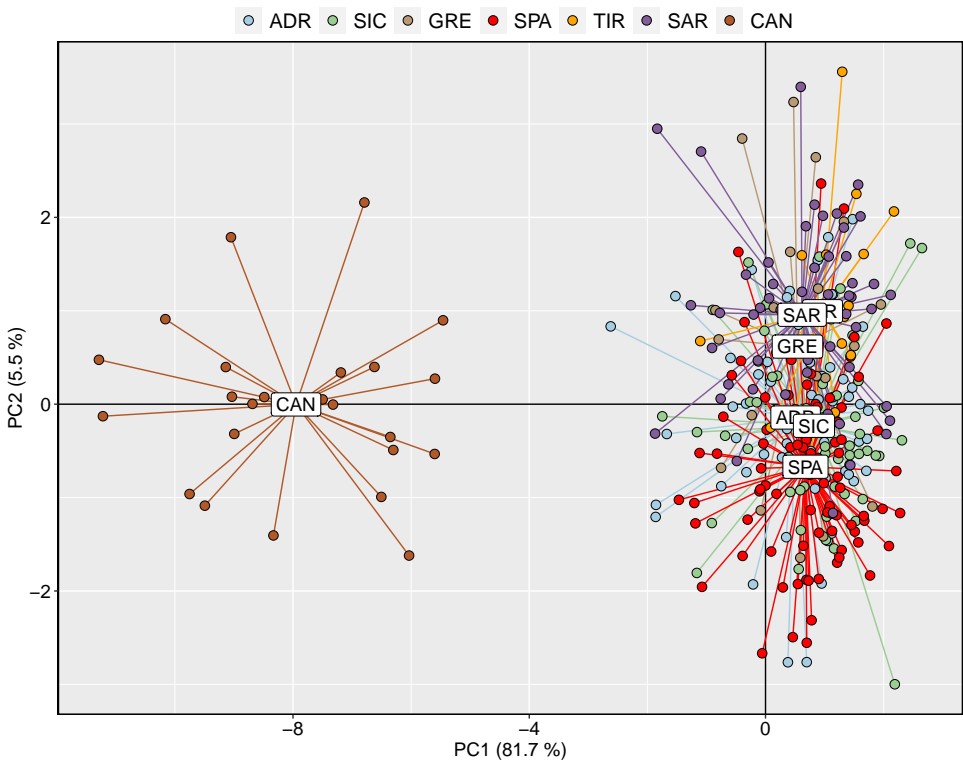

**Figure 3 Discriminant analysis of principal components (DAPC) with a priori geographical information.** Scatterplot of the discriminant analysis of principal components of the microsatellite data for 7 locations. Individual genotypes appear as dots. Each population is indicated by abbreviation reported in Table 1.

DAPC analysis performed excluding the Canadian sample corroborated the previous result identifying $K = 3$ as the most probable number of clusters within the Mediterranean Sea (Fig. 5). All pairwise $F_{ST}$ comparisons among clusters detected by the DAPC were statistically significant. Higher values were reported comparing Cluster 4 (Atlantic samples) among Mediterranean clusters: 1 ($F_{ST} = 0.13$), 2 ($F_{ST} = 0.09$), and 3 ($F_{ST} = 0.11$). Among Mediterranean genetic groups, pairwise $F_{ST}$ ranged from 0.05 (cluster 1–cluster 3) to 0.07 (cluster 2–cluster 3).

## DISCUSSION

This study aimed to investigate the genetic structure of the swordfish *Xiphias gladius*, within the Mediterranean Sea and between NW-Atlantic and Mediterranean populations using a panel of 18 polymorphic microsatellite loci. These results indicate clear inter-oceanic genetic differentiation among NW-Atlantic and Mediterranean stocks with an improved ability to assign individuals to their population of origin compared to previous multi-locus works (*Kotoulas et al., 2007*; *Smith et al., 2015*). Secondly, conversely to previous studies, the multivariate analysis based on microsatellite dataset suggested the presence of three mixed genetic groups within the Mediterranean Sea.
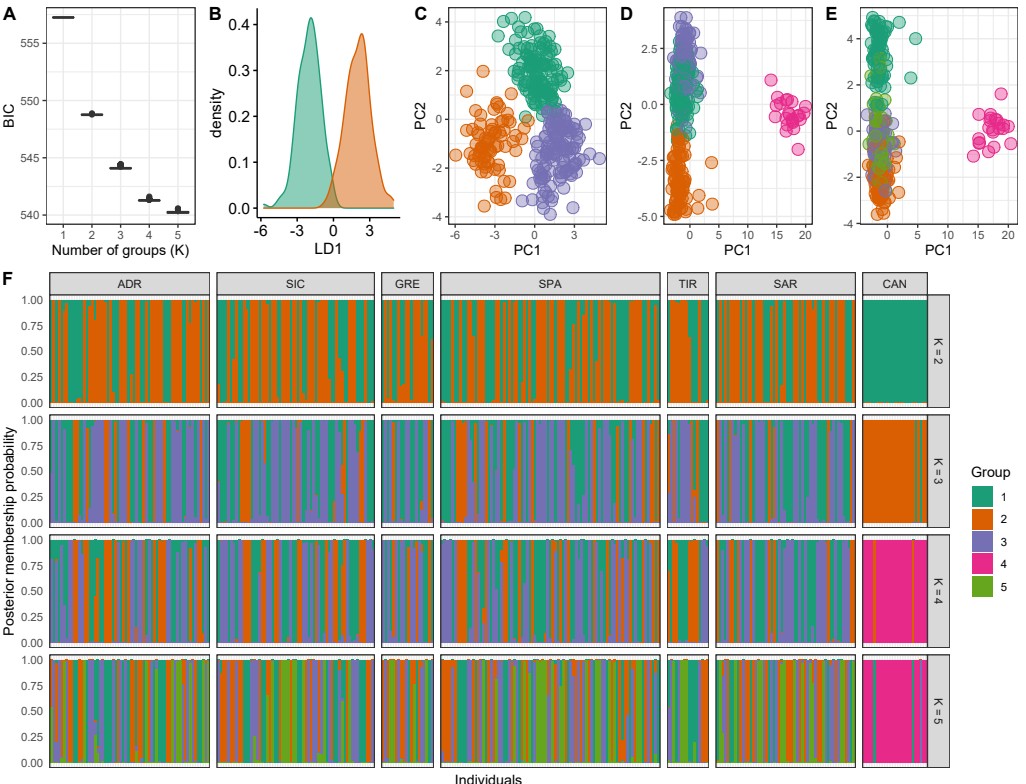

**Figure 4   Discriminant analysis of principal components (DAPC) results including Canadian sample.** (A) The optimal number of clusters (K) as determined by 'k-means'. (B–E) Scatterplots based on the DAPC output for K from 2 to 5. Dots represent different individuals and colours represent different clusters. (F) Barplots showing the probabilities of assignment of individuals to K from 2 to 5 genetic DAPC clusters. Each individual is represented as a vertical bar, with colours corresponding to probabilities of membership into the clusters.

Before examining the results, we must address the mains caveat of this study: sampling sites. The samples for this study were collected opportunistically. Therefore we cannot evaluate hypothesis related to putative homing of the species towards breeding areas or to evaluate admixture in the feeding locations.

The Mediterranean population showed a lower level of genetic variability compared to Atlantic ones, corroborating the outcomes of previous studies based on microsatellites (*Reeb, Arcangeli & Block, 2003*; *Kotoulas et al., 2007*; *Kasapidis et al., 2009*) and mtDNA (*Bremer, Baker & Mejuto, 1995*; *Bremer et al., 1996*; *Bremer et al., 2005b*; *Kotoulas et al., 1995*; *Rosel & Block, 1996*). The low genetic variability and small effective population size of the Mediterranean swordfish population could be a consequence of the semi-enclosed nature and the limited size of the Mediterranean basin (*Bremer et al., 2005b*; *Kasapidis et al., 2007*).

The dataset allowed to clearly identify a genetic structure between the Mediterranean and Atlantic populations according to previous studies (*Bremer, Baker & Mejuto, 1995*; *Kotoulas et al., 1995*; *Kotoulas et al., 2007*; *Bremer et al., 1996*; *Rosel & Block, 1996*; *Pujolar,*

Peer**J**

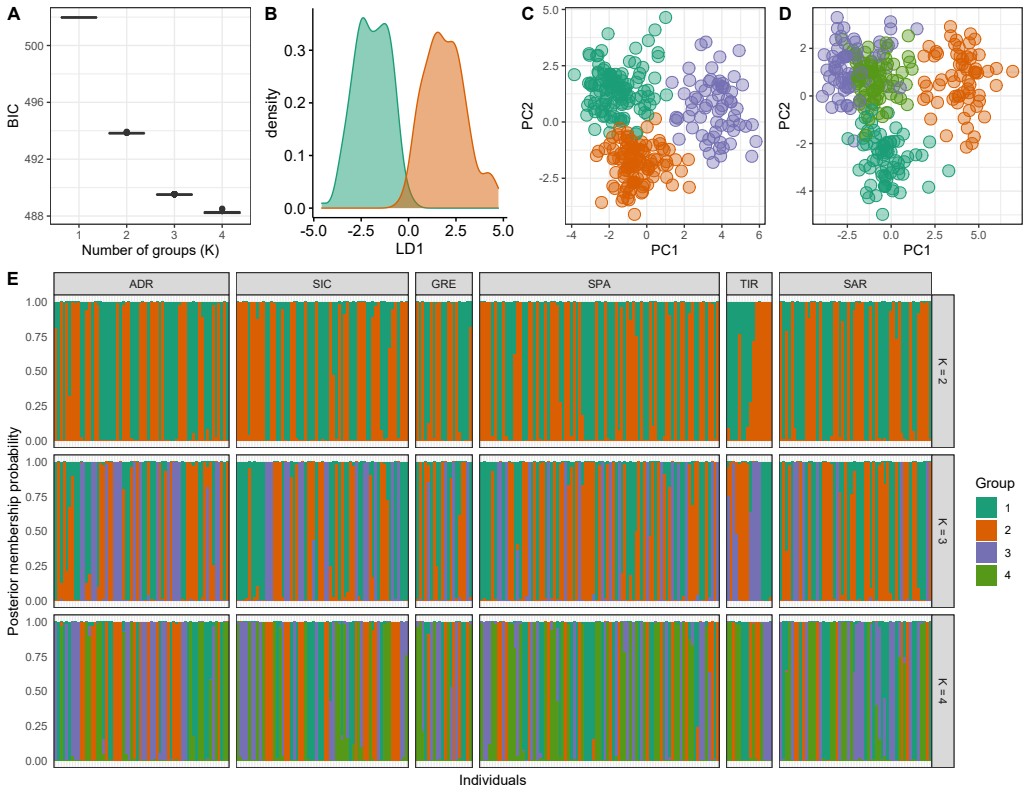

**Figure 5 Discriminant analysis of principal components (DAPC) results excluding the Canadian sample.** (A) The optimal number of clusters (K) as determined by 'k-means'. (B–D) Scatterplots based on the DAPC output for K from 2 to 4. Dots represent different individuals and colours represent different clusters. (E) Barplots showing the probabilities of assignment of individuals to K from 2 to 4 genetic DAPC clusters. Each individual is represented as a vertical bar, with colours corresponding to probabilities of membership into the clusters.

*Roldan & Pla, 2002*; *Viñas et al., 2007*; *Smith et al., 2015*). The presence of these populations is supported by pairwise $F_{ST}$, multivariate analysis (DAPC) and Bayesian genetic clustering using STRUCTURE. However, the estimates of differentiation ($F_{ST}$) calculated in this work between the two stocks (0.091), is higher than those reported by recent works also using a multi-locus approach (*Kotoulas et al., 2007*; *Smith et al., 2015*). A low $F_{ST}$ value ($F_{ST} < 0.03$) was reported between Atlantic and Mediterranean swordfish stocks using four highly polymorphic microsatellites (*Kotoulas et al., 2007*), while $F_{ST}$ values around 0.07 were reported by *Smith et al. (2015)* analysing 26 single nucleotide polymorphisms (SNPs) within 10 nuclear genes. Moreover, *Smith et al. (2015)*, by using this approach, effectively distinguished the population of the North Atlantic from that of the South Atlantic unlike previous efforts based on 4 microsatellites (*Kasapidis et al., 2007*; *Kotoulas et al., 2007*). The greater $F_{ST}$ observed in this study likely results from the greater number of loci analysed, which could enhance the resolution and improve the precision of estimates of genetic distance (*Nei, 1978*; *Kalinowski, 2005*).

Although both STRUCTURE and DAPC identify interoceanic genetic differentiation, the results obtained from STRUCTURE clustering methods using microsatellites were not concordant, with DAPC providing evidence. DAPC, in fact, provided evidence of a substructure inside the Mediterranean Sea. The discrepancy between clustering methods may be attributable to the model-based methods employed. Bayesian methods typically fail to identify some complex types of spatial structure such as isolation-by-distance (*Jombart, Devillard & Balloux, 2010*) and hierarchical population structure (*Evanno, Regnaut & Goudet, 2005*), and fail to detect any genetic structure when genetic divergence is very low ($F_{ST} < 0.03$) (*Latch et al., 2006*; *Waples & Gaggiotti, 2006*; *Duchesne & Turgeon, 2012*). On the other hand, the ability of DAPC to identify genetic clusters when STRUCTURE failed was reported by *Jombart, Devillard & Balloux (2010)* and *Kanno, Vokoun & Letcher (2011)*. Thus, DAPC can outperform the STRUCTURE method in inferring the number of subpopulations when they are weakly differentiated as in our study case.

In the present study, the Mediterranean swordfish display a week but significant structure where clusters, detected by DAPC, are spatially admixed with no geographic pattern; a result corroborated by pairwise $F_{ST}$ estimation. Population structure within the Mediterranean Sea is not consistent with previous work based on microsatellite data (*Kotoulas et al., 2007*). However, *Kotoulas et al. (2007)* analysed only four microsatellites; a very low number of loci with which to attempt to detect the presence of genetic differentiation, especially in the case of pelagic fish species. Genetic homogeneity across Mediterranean sampling localities was, also, observed by previous studies based on allozymes data (*Pujolar, Roldan & Pla, 2002*), RFLPs of the entire mtDNA (*Chow et al., 1997*), analyses of a single-copy nuclear calmodulin gene and PCR–RFLP data of the mtDNA CR (*Chow & Takeyama, 2000*). However, sample homogeneity does not necessarily equate to population homogeneity (*Ward, 2000*) and population differentiation may be obscured by population mixture in wintering or feeding areas, especially for highly migratory species (Van (*Wagner & Baker, 1990*; *Bowen et al., 1992*; *Wenink, Baker & Tilanus, 1993*; *Bremer et al., 2005a*). Swordfish is a highly migratory species, and in the Atlantic Ocean, it is able to cover annually very long distances as reported by pop-up satellite archival tags analysis (*Neilson et al., 2014*; *Abascal et al., 2015*). The same ability was also observed within the Mediterranean Sea (*Canese et al., 2008*). Evidence of shared areas has been observed in the feeding area west of the Strait of Gibraltar where Mediterranean and North Atlantic swordfish populations mix (*Bremer et al., 2005a*; *Viñas et al., 2007*; *Smith et al., 2015*). Furthermore, admixture between North and South Atlantic populations occurs over a broader geographic area from Western Sahara to the Iberian sea extending west towards the central North Atlantic and then south towards the northern Brazilian coast (*Smith et al., 2015*). Considering each of the Mediterranean sampling areas we note that no evidence of spawning activities has been observed in the Adriatic Sea, (*Arocha, 2007*) and it is unlikely that a mixture of individuals originating from different subpopulations would ever result as a consequence of a transitional effect due to its semi-enclosed sea characteristic. Rather, individual admixture in the Adriatic Sea is likely a consequence of its use as a feeding ground. As for the other sampling areas, they could represent transitional zones with mixing between populations as occurs between North and South Atlantic swordfish. The hypothesis of an admixed population within the

**Table 4  Descriptive statistics for each location over all loci.**

| Location | n | MNA (s.d) | AR | H$_O$ (s.d) | H$_E$ (s.d) | HW disequilibrium | F$_{IS}$ (s.e) | Null alleles (s.d) |
|---|---|---|---|---|---|---|---|---|
| ADR | 62 | 5.556 (3.166) | 4.569 | 0.543 (0.253) | 0.560 (0.254) | n.s. | 0.027 (0.077) | 0.043 (0.058) |
| SIC | 61 | 5.444 (3.072) | 4.474 | 0.505 (0.247) | 0.558 (0.242) | Significant | 0.086 (0.169) | 0.058 (0.096) |
| GRE | 20 | 4.875 (1.928) | 4.278 | 0.552 (0.120) | 0.609 (0.120) | n.s. | 0.096 (0.124) | 0.049 (0.062) |
| SPA | 85 | 6.062 (3.492) | 4.453 | 0.571 (0.180) | 0.621 (0.175) | Significant | 0.077 (0.138) | 0.059 (0.058) |
| TIR | 16 | 4.706 (2.114) | 4.443 | 0.554 (0.223) | 0.583 (0.205) | n.s. | 0.037 (0.217) | 0.042 (0.056) |
| SAR | 54 | 5.706 (3.016) | 4.495 | 0.503 (0.229) | 0.574 (0.219) | Significant | 0.126 (0.156) | 0.073 (0.091) |
| CAN | 25 | 8.938 (4.389) | 7.048 | 0.715 (0.119) | 0.770 (0.153) | Significant | 0.059 (0.109) | 0.078 (0.104) |

Notes.

n, sample size; MNA, mean number of alleles; AR, allelic richness; H$_o$, observed heterozygosity; H$_E$, expected heterozygosity; HW, disequilibrium, significance for the HardyWeinberg disequilibrium after Bonferroni correction; F$_{IS}$, mean null allele frequencies; s.d. and s.e, standard deviation and standard error.

Mediterranean Sea is also supported by the significant excess of homozygote genotypes detected in Mediterranean samples (Table 4 and Table S1). An excess of homozygotes may be due to genotyping errors such as null alleles, allele dropout and stuttering, or it can be a consequence of inappropriate sample size. However, a biological explanation for the occurrence of Hardy Weinberg disequilibrium (HWD) is known as the Wahlund effect (WE). According to the WE, HWD can appear to occur when the sample analysed is composed of a mix of distinct subpopulations, as would be expected by highly migratory and spatially structured species.

Philopatric behaviour has been identified as the driving force behind the structuring of very high migratory pelagic fish *Istiompax indica* (Williams et al., 2016), *Gadus morhua* (Bonanomi et al., 2016) and *Thunnus thynnus* (Rooker et al., 2008; Aranda et al, 2013). In swordfish, spawning site fidelity is supported by both high levels of genetic differentiation obtained comparing separated breeding areas in the Atlantic Ocean (Bremer et al., 2005a; Bremer et al., 2007) and the evidence of the mixing areas with minimal gene flow between Mediterranean and North Atlantic swordfish populations (Bremer et al., 2007; Viñas et al., 2007; Smith et al., 2015). Seasonal site (foraging and spawning) fidelity was also suggested within the Mediterranean Sea by the recapture of tagged individuals, that generally occurred in the same area of tagging also after several years (Garibaldi & Lanteri, 2017). Philopatric instinct was already suggested by Viñas et al. (2010) as a possible cause of swordfish population differentiation within the Mediterranean Sea. The authors suggested population substructure based on a clinal decrease in mitochondrial genetic variability from the western to the eastern basins. They hypothesised that population differentiation may be the consequence of distinct phylogeographic histories of populations in the Eastern and the Western Mediterranean basins and is maintained by present-day life-history traits, including homing fidelity to spawning sites. Within the Mediterranean Sea, three main spawning areas are currently recognized. The first one is located in the western part of the basin, the second one in the eastern part and the third one extends from the southern Tyrrhenian Sea to the Ionian Sea (Cavallaro, Potoschi & Cefali, 1991; Tserpes, Peristeraki & Somarakis, 2001; Arocha, 2007; Tserpes, Peristeraki & Valavanis, 2008). Although there is no evidence that these spawning aggregations represent discrete stocks, geographical

localization of these three discrete spawning areas may explain the three genetic clusters observed in this study. However, the sampling design used in this study does not allow us to test this hypothesis.

Rejection of a model of panmixia in a relatively small sea basin is not new for large pelagic species. For example, for the Atlantic bluefin tuna (ABFT), which shares the same spawning grounds as swordfish (*Bonales et al., 2019*), a remarkable homing behaviour to Tyrrhenian and Ionian spawning grounds has been detected by satellite tracks (*Reeb, 2010*). Reproductive isolation explains a fine-scale structure identified among Balearic, Tyrrhenian, and Ionian juvenile bluefin tuna using both nuclear microsatellite loci and the mtDNA control region (*Carlsson et al., 2004*). Furthermore, spatially and temporally stable genetic structure was observed between Adriatic and Tyrrhenian bluefin tuna (*Riccioni et al., 2010*).

## CONCLUSIONS

The present study suggests genetic heterogeneity within the Mediterranean Sea swordfish stock supporting previous study (*Viñas et al., 2010*). Despite the low differentiation observed, these results provide useful information on the stock structure of the swordfish, contributing evidence for the rejection of the hypothesis for a single Mediterranean population. The high degree of separation between Mediterranean spawning areas of swordfish towards these areas could support a weak degree of genetic differentiation. Although weakly differentiated, the presence of genetically distinct clusters warrants reconsidering the appropriateness of the current single-stock approach used by ICCAT. Management recommendations and measures which assume stock uniformity across large regions may result in overfishing of some small, but discrete demographic units. However, the high level of mixing at sample locations hampers a clear delineation into corresponding spatial management areas. In light of these results, further investigations are required to determine the degree of complexity of the Mediterranean swordfish population structure to achieve effective swordfish conservation. To date, significant gaps still exist regarding Mediterranean swordfish life history and stock structure. Including in the analysis larvae and young-of-the-year (YOY) would assist in the assessment of swordfish population dynamics. Collecting a larger sample size from each spawning grounds, maximizing stock discreteness could ameliorate the evaluation of geographic genetic segregation. Moreover, including tagging information, currently very limited for Mediterranean swordfish, would be necessary to better resolve the swordfish migratory behaviour.

## ACKNOWLEDGEMENTS

We truly thank the fishermen, Pesca Pronta Import-Export s.r.l. for their extremely precious collaboration during sampling activities. The authors wish to thank all the staff at OCEANIS s.r.l. (Ercolano, Naples) and Dr Alessandro Lucchetti (ISMAR-CNR, Ancona) for the support in sampling activities. We also thank Tom Jenkins ( https://github.com/Tom-Jenkins) and BJ Knaus, and NJ Grünwald (https://grunwaldlab.

github.io/Population_Genetics_in_R/clustering_plot.html) for the R scripts for graphical results representation.

### Funding
This work was funded by the Ministry of Agriculture, Food and Forestry Policies (MIPAAF), note 6775, Art.36 Paragraph 1 Reg (UE9 n 508/2014) to Oliana Carnevali. The funders had no role in study design, data collection and analysis, decision to publish, or preparation of the manuscript.

### Grant Disclosures
The following grant information was disclosed by the authors:
Ministry of Agriculture, Food and Forestry Policies: UE9 n 508/2014.

### Competing Interests
The authors declare there are no competing interests.

### Author Contributions
- Tommaso Righi conceived and designed the experiments, performed the experiments, analyzed the data, prepared figures and/or tables, authored or reviewed drafts of the paper, and approved the final draft.
- Andrea Splendiani conceived and designed the experiments, performed the experiments, authored or reviewed drafts of the paper, and approved the final draft.
- Tatiana Fioravanti performed the experiments, authored or reviewed drafts of the paper, and approved the final draft.
- Andrea Petetta performed the experiments, analyzed the data, prepared figures and/or tables, authored or reviewed drafts of the paper, and approved the final draft.
- Michela Candelma, Giorgia Gioacchini, Kyle Gillespie, Alex Hanke, Oliana Carnevali and Vincenzo Caputo Barucchi conceived and designed the experiments, authored or reviewed drafts of the paper, and approved the final draft.

### Data Availability
The raw measurements are available in Data S1.

### Supplemental Information
Supplemental information for this article can be found online at http://dx.doi.org/10.7717/peerj.9518#supplemental-information.

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
