# Peer review of "Mediterranean swordfish (Xiphias gladius Linnaeus, 1758) population structure revealed by microsatellite DNA: genetic diversity masked by population mixing in shared areas"

_PeerJ, doi:10.7717/peerj.9518_

## Round 0.1 · original submission · Major Revisions

Dear Authors,

Thank you for your submission to PeerJ.I have now received two reviews of your manuscript, and both recommended major revisions.

The reviewers provided detailed comments and suggestions on how to improve the article, and I would like you to follow these suggestions. When you send in a revision, if you choose to resubmit this MS, please provide a detailed response to the reviewer’s comments as well.

In my own opinion, as presented, the article has a number of major weaknesses, many of them addressed by the reviewers, but the main ones I am concerned with is the lack of description of the sampling design and its appropriateness to the stated aims. I am well aware that often sampling is opportunistic, but that means that the objectives need to be tailored more carefully.

I look forward to seeing a revised version of this MS in the near future.

Rita Castilho

Reviewer 1 ·

Basic reporting

Revision of the manuscript entitled “Mediterranean swordfish (Xiphias gladius Linnaeus, 1758) population structure revealed by microsatellite DNA: genetic diversity masked by population mixing in shared areas” by Righi et al

The manuscript asses the genetic population structure of swordfish from 6 locations within the Mediterranean and one location in the Atlantic (west Atlantic, Canadian coast) This is probably the most exhaustive analysis of this species in this Sea, analyzing up to 300 individuals with 20 microsatellite loci. The manuscript is well written, although I am not an English native speaker, the English sounds good to me.
Literature references are up to date, and the article has a correct structure.


The main finding of the manuscript is the possible evidence of population structure within the Mediterranean. However, this genetic structure has not any geographical nor biological basis. Thus, the authors should implement several improvements before publication.

1.- My major concern is about the detection of the genetic structure, based on the DAPC results, with the presence of three genetic clusters. These clusters are not related to any geographical nor biological trait. The authors hypothesize the relation of these genetic clusters to a behavior of homing towards breeding areas. However, the sampling scheme used in this study is not compatible for evaluating this hypothesis. Some specification of the sampling used is needed. For instance, are some of the locations sampled correspond to spawning or feeding grounds. Are the individual’s juveniles or adults? It can be understood that the availability of samples is not always the desired, but better sampling description and a discussion of results based on the current sampling should be done.
2.- Clear genetic differentiation is observed between Canadian location and the Mediterranean. This is not surprising, similar results have been published in multiples studies of swordfish. I wonder, about the distribution of locations in each DAPC cluster. For instance, one question interesting to ask, is the level of admixture in the feeding locations, which can be probably higher than the spawning locations. But, without accurate sampling descriptions very difficult to answer these questions.
3.- Better discussion, some authors using a similar sampling revealed an incipient genetic differentiation within the Mediterranean. A better description of this fact with the comparisons with the current results would improve the manuscript.

3.- Several minor comments:
Abstract.
Line 28. “for the first time” this is not accurate. At least one previous study of Microsatellite in the Mediterranean is already published (Kotoulas 2007).
Materia and methods.
Line 168 to 198. These two paragraphs should be reordered. Please first specify all the parameters and software used for STRCUTURE analysis. Then the parameters and specifications of DAPC analysis.
Table 1. better description of the sampling with possible identification of samples from breeding and spawning areas. Please consider to include a Map.
Table 2. Could you provide the range size of each loci.
Table 3 and 4. have you done multiple test correction?
Figure 2C. I believe that the last location should be labelled as CAN.

Experimental design

As mentioned before, the main problem of this study is that the sampling design is no adequate for the testing the hypothesis of admixture within the Mediterranean. There are no basal populations and, thus, very difficult to describe the admixture between possible different populations. At least a better description of the sampling should be included

Validity of the findings

It is difficult to validate the results since the sampling is adequate. Under my point of view the authors should be more restricted about their conclusions of admixture within the Mediterranean.

Additional comments

All general comments are already include int he first part of the revision.

Reviewer 2 ·

Basic reporting

This is a quite interesting work, that try to fill the gap about this species in the Mediterranean Sea.
The authors followed the PeerJ instructions for authors, but the manuscript need for a revision of the English (both typos and structure). The English language should be improved to ensure that an
international audience can clearly understand your text. The figures, especially the barplots are not of high quality. It’s difficult to read the population names. Also, in the Fig 2C there is a labelling error with two groups labelled as ADR of which one should be CAN. Some references are not in alphabetical order, and a revision of this part is needed (please see comments). RAW data are supplied and clear.

Experimental design

The experimental design of this work is what worry me the most. The questions are clear and well defined in the text, but the sampling design is not really clear, and also some methods need to be revised. It is not mentioned the stage (or size, or sex stage) of the specimens used for the analyses, and this is of vital importance for similar works on pelagic species. The authors used to sample just during the spawning months, and apparently, not juveniles were used for this analysis. I think this is a critical point, since the species is well known to be extremely vagile among spawning areas. The research is very useful to better investigate the Mediterranean-Atlantic genetic structure of this species, but the flaws on the sampling design and "consequently" methods are preventing a reliable final result.

Validity of the findings

I would say that the findings are useful, but clarifications about the sampling design and some analysis (mainly the DAPC and related Fst) are needed before Acceptance (see comments). This is very critic. I think this is the weakest part of the work, which generally I found very useful.

Additional comments

Abstract:

Your methods in the abstract needs to be restructured in order to be more concise and without general information about the utility of the microsatellite, which is not requested here. Try to change in something like:
“For the first time, microsatellite DNA was used to investigate fine-scale population structuring of swordfish from six main fishing areas of the Mediterranean Sea, in order to test the uncertainty of the stock delineation of the species in this area.”

Introduction:

Line 40-41: change “Sustainability in harvesting fish stocks to avoid their depletion and to guarantee survival and long-term production is one of the primary objectives of fishery management (Cochrane, 2002).” to “Sustainability in harvesting fish stocks, to avoid their depletion and to guarantee survival and long-term production, is one of the primary objectives of fishery management (Cochrane, 2002).”

Line 42: More than “isolated” they are “structured”. Not necessarily isolated.

Line 49: Delete “commercial”

Line 54-59: This part needs to be restructured. It seems you are talking about your results, and not the results from cited works.

Line 63-64: The Reference for this is incomplete. Please use the correct IUCN reference: “Collette, B., Acero, A., Amorim, A.F., Bizsel, K., Boustany, A., Canales Ramirez, C., Cardenas, G., Carpenter, K.E., de Oliveira Leite Jr., N., Di Natale, A., Die, D., Fox, W., Fredou, F.L., Graves, J., Guzman-Mora, A., Viera Hazin, F.H., Hinton, M., Juan Jorda, M., Minte Vera, C., Miyabe, N., Montano Cruz, R., Masuti, E., Nelson, R., Oxenford, H., Restrepo, V., Salas, E., Schaefer, K., Schratwieser, J., Serra, R., Sun, C., Teixeira Lessa, R.P., Pires Ferreira Travassos, P.E., Uozumi, Y. & Yanez, E. 2011. Xiphias gladius (errata version published in 2016). The IUCN Red List of Threatened Species 2011: e.T23148A88828055. https://dx.doi.org/10.2305/IUCN.UK.2011-2.RLTS.T23148A9422329.en. Downloaded on XXX”

Line 90-92: Smith et al. (2015) (https://doi.org/10.1371/journal.pone.0127979) have shown, even if very low, a level of gene flow between North Atlantic and the Mediterranean Sea.

Line 108-109: I would suggest to remove “for the first time”. Despite the larger number of microsatellites used in the present study, it seems that a multilocus approach have already been used, even if with just 4 microsatellites (Kotoulas et al., 2007).

MATERIALS & METHODS

Line 121: The samples were collected between May and October, during the intensive spawning for this species in the Mediterranean. The spawning seems to be unrecorded during December and January, a period rich in Juveniles. I’m wondering how much this sampling design may have masked a potential genetic structure, masked by the genetic characterization of large and migratory adults, rather than genotype less vagile juveniles. Also, I don’t see any information about the size of the specimen sampled. Were they all adults or both adults and juveniles?

Line 123: Remove “was”

Line 124: Change “for accordance” into “in accordance”.

Line 127: Change “following its specific protocol” into “following the manufacturers' protocols.”

Line 130: Change “Microsatellite amplification and genotyping and diversity analysis” into “Microsatellite amplification, genotyping and diversity analysis”.

Line 135: Change “PCA amplifications” into “The PCR amplifications”.

Line 136: Change “PCA amplifications” into “The PCR amplifications”.

Line 139: How was the amount of DNA measured?

Line 144: Change “appropriate” into “specific”

Line 153-155: This is not very clear to me. I would suggest to specify better this step, avoiding colloquial words such “other errors”.

Line 172: Change “employed” into “used”

Line 168-177: I would focus more on the different approach of both methods. For example, the DAPC doesn’t rely on a priori population assignment, but it would ideally need an a priori number of clusters.

Line 179-180: Change “and hierarchical population structure (Evanno, Regnaut & Goudet,
2005) and correct allocation…” into “and hierarchical population structure (Evanno, Regnaut & Goudet, 2005), and correct allocation…”.

Line 181-184: I would suggest to better explain this in the STRUCTURE and DAPC paragraph, explaining why (different models and math behind).

Line 191: The R package should be mentioned above, in the clustering method paragraph.

RESULTS
Line 219-220: I would like to suggest the reading of this manuscript about the effect of NA on the estimation of differentiation https://doi.org/10.1093/molbev/msl191

Line 231-232: This is a bit in contrast with the effect of NA on the Fst, for which when a strong Fst is observed, the effect of NA on the Fst values is heavier. https://doi.org/10.1093/molbev/msl191
How the authors explain that?

Line 234: Change "The lowest and no statistically significant" into "Low and statistically not significant".

Line 236: Change "highest" into "higher".

Line 246-250: This additional analysis is not mentioned in the Methods. Why?

DISCUSSIONS:

Line 266-268: I think a citation is missing here. Improved how and compared to what?

Line 269-270: I’m not sure the analyses are suggesting three genetic groups. The problem is on the definition of cluster during the analysis (especially the pre-DAPC). Looking at the DAPC plot, the clusters within the Mediterranean are just two. Even the authors of the method (DAPC) suggest caution on this.

Line 294-297: I would focus also on the different nature of the analysis. It is more about the choose of clusters to use in the analysis of DAPC.

Line 297-299: The authors highlighted that the clusters found are not separated geographically, so I think this is not the case. Also, I think this is more related to the stage of individuals. Are all migrant adults? This may explain some of the results. I think the authors should consider this. I’m still wondering how the sampling design affected the results (adult spawners vs. juveniles).

Line 299-302: I suggest the authors to be cautious about that.

Line 371-371: This assumption needs more analysis, since the formation of more than 1 cluster within the Mediterranean have been observed mainly on the assumptions of the best K methods used, which are very sensitive to bias.

Line 381-383: I would suggest the authors to avoid to consider the genetic clusters observed here as “discrete populations”.

Line 386-388: This is the only clue about the use or not of juveniles.

Line 389: I wouldn’t use to collect just during spawning. It causes mixing situation.



REFERENCES:
Please check carefully that everything is correct and in line with the instructions for authors. I’ve noticed some weird non-alphabetically order in the references. E.g: Kasapidis et al. between two Kotoulas et al.

Kotoulas G, Magoulas A. 2007. Genetic structure of the swordfish (Xiphias gladius) stocks in the Atlantic using microsatellite DNA analysis. Collect. Vol. Sci. Pap. ICCAT 61:89–98.

Kasapidis P, Pakaki V, Kotoulas G, Magoulas A. 2009. Isolation and characterization of 18 new polymorphic microsatellite loci for the swordfish, Xiphias gladius. Molecular ecology resources 9:1383–1386.

Kotoulas G, Magoulas A, Tsimenides N, Zouros E. 1995. Marked mitochondrial DNA differences between Mediterranean and Atlantic populations of the swordfish, Xiphias gladius. Molecular Ecology 4:473–482.

Kotoulas G, Mejuto J, Antoniou A, Kasapidis P, Tserpes G, Piccinetti C, Peristeraki P, Garcia-Cortes B, Oikonomaki K, De la Serna JM. 2007. Global genetic structure of swordfish (Xiphias gladius) as revealed by microsatellite DNA markers. ICCAT Coll Vol Sci Pap 61:79–88.

---

## Round 0.2 · Major Revisions

Dear authors,

I have received your resubmission, and on the overall, you addressed the reviewers' comments passably. However, some points remained inadequately dealt with.

My opinion is that before sending the manuscript out for another round of reviews, I would like you to take my comments on board and make whichever changes you see fit (please check the scripts at the end of the pdf file I am attaching). I think you could have invested a bit more time to illustrate your results and claims so that reviewers and hopefully the readers can fully appreciate your paper. Overall, I maintain that the MS and the data have the potential to be an important contribution to the understanding of population structuring in an economically important fishery species. However, as I said, the MS still needs improvements, and thus I recommend major revisions again.

I look forward to seeing a revised version of this MS in the near future.

---

## Round 0.3 · Minor Revisions

First, I would like to express my appreciation for your re-submission and for the ever so timely completion of the changes which is duly noted.

The revisions addressed the major concerns of the referees and my own to my satisfaction. I do not see the need to send the manuscript out for reviewing again. However, I still have a couple of comments and therefore I am returning this version for minor revisions.

POINT 1. Regarding figure 1, please insert the name of the locations next to the locations points, in that way the reader immediately will see the ID of the locations instead of having to navigate between colours. Also, I would advise for a more neutral colour for the map background.

POINT 2. Regarding the lack of an overview of results. I suggest adding a table with the descriptive statistics for each location over all loci (location, code, longitude, latitude, number of individuals, mean number of alleles, allelic richness, observed heterozygosity, expected heterozygosity, significance of Hardy-Weinberg equilibrium) and another table for descriptive statistics for the microsatellite loci over the locations (loci designation, number of individuals, total allele number, mean number of alleles, allelic richness, observed heterozygosity, expected heterozygosity, HWE test result; inbreeding coefficient; null allele frequency (see the example from tables 1 and 2 from Zarraonaindia I, Pardo MA, Iriondo M, Manzano C, Estonba A 2009. Microsatellite variability in European anchovy (Engraulis encrasicolus) calls for further investigation of its genetic structure and biogeography. ICES Journal of Marine Science 66:1–7.

I will look forward to your resubmission so that we can move the manuscript to the production stage!

---

## Round 0.4 · accepted · Accept

Dear Dr.Righi,

The revisions of this paper feature many improvements over the original submission, thank you for making the suggested edits. Congratulations on the work and thanks for choosing PeerJ.

I am happy to accept it now and move it forward into production.